# *Legionella longbeachae* effector protein RavZ inhibits autophagy and regulates phagosome ubiquitination during infection

Yunjia Shi[1☯], Hongtao Liu[1☯], Kelong Ma[1], Zhao-Qing Luo[2]*, Jiazhang Qiu[1]*

**1** State Key Laboratory for Zoonotic Diseases, Key Laboratory for Zoonosis Research of the Ministry of Education, College of Veterinary Medicine, Jilin University, Changchun, China, **2** Purdue Institute for Inflammation, Immunology and Infectious Disease and Department of Biological Sciences, Purdue University, West Lafayette, Indiana, United States of America

☯ These authors contributed equally to this work.
* qiujz@jlu.edu.cn (JQ); luoz@purdue.edu (ZQL)

**Data Availability Statement:** All relevant data are within the paper and its Supporting information files.

## Abstract

*Legionella* organisms are ubiquitous environmental bacteria that are responsible for human Legionnaires' disease, a fatal form of severe pneumonia. These bacteria replicate intracellularly in a wide spectrum of host cells within a distinct compartment termed the *Legionella*-containing vacuole (LCV). Effector proteins translocated by the Dot/Icm apparatus extensively modulate host cellular functions to aid in the biogenesis of the LCV and intracellular proliferation. RavZ is an *L. pneumophila* effector that functions as a cysteine protease to hydrolyze lipidated LC3, thereby compromising the host autophagic response to bacterial infection. In this study, we characterized the RavZ (RavZ$_{LP}$) ortholog in *L. longbeachae* (RavZ$_{LLO}$), the second leading cause of *Legionella* infections in the world. RavZ$_{LLO}$ and RavZ$_{LP}$ share approximately 60% sequence identity and a conserved His-Asp-Cys catalytic triad. RavZ$_{LLO}$ is recognized by the Dot/Icm systems of both *L. pneumophila* and *L. longbeachae*. Upon translocation into the host, it suppresses autophagy signaling in cells challenged with both species, indicating the functional redundancy of RavZ$_{LLO}$ and RavZ$_{LP}$. Additionally, ectopic expression of RavZ$_{LLO}$ but not RavZ$_{LP}$ in mammalian cells reduces the levels of cellular polyubiquitinated and polyneddylated proteins. Consistent with this process, RavZ$_{LLO}$ regulates the accumulation of polyubiquitinated species on the LCV during *L. longbeachae* infection.

## Introduction

*Legionella* are gram-negative bacteria ubiquitously found in natural environments [1]. When contaminated aerosols are inhaled by humans, *Legionella* can cause a severe form of pneumonia termed Legionnaires' disease, which can be fatal if not promptly and appropriately treated [1]. To date, more than 60 *Legionella* species have been recognized, and almost half of them have been reported to cause human disease [2]. *Legionella pneumophila* and *Legionella longbeachae* are the most frequently isolated species in cases of Legionnaires' disease [2]. *L.*

**Funding:** This work was supported by National Natural Science Foundation of China (grant #: 31970134 and 32170182 to Jiazhang Qiu); the Thousand Young Talents Program of the Chinese government (Jiazhang Qiu) and startup fund from Jilin University and the First Hospital of Jilin University. The funders had no role in study design, data collection and analysis, decision to publish, or preparation of the manuscript.

**Competing interests:** The authors have declared that no competing interests exist.

*pneumophila* accounts for up to 90% of clinical disease worldwide [3]. However, infections resulting from *L. longbeachae* were reported as often as those resulting from *L. pneumophila* in Australia and New Zealand [3]. In addition, during the past decade, the incidence of *L. longbeachae* infection has increased markedly across regions of Europe and Asia [2]. The clinical presentations of *L. pneumophila* and *L. longbeachae* infections are similar. However, these two species exhibit distinct features in their environmental niches, transmission routes, and epidemiology [4]. *L. pneumophila* is primarily found in artificial and natural aquatic environments [5], whereas *L. longbeachae* is mainly associated with compost and potting soils [6]. The adoption of distinct ecological niches is determined by differences in the physiological and genetic features between the two species. For instance, *L. longbeachae* is encapsulated and encodes genes to produce proteins that are involved in the degradation of plant cell walls [4]. Unlike *L. pneumophila*, *L. longbeachae* does not encode flagella biosynthesis genes, thus partly explaining the differences in mouse susceptibility between these two species [4, 7]. Additionally, despite the presence of homologs of the regulatory genes that regulate the *L. pneumophila* biphasic lifestyle, global gene expression analysis revealed a significantly less pronounced life cycle switch in *L. longbeachae* [4].

*Legionella* spp. replicates intracellularly in a wide range of host cells, including free-living protozoa and alveolar macrophages, in a membrane-enclosed compartment called the <u>Legio</u>nella-<u>c</u>ontaining <u>v</u>acuole (LCV). Upon phagocytosis by host cells, phagosomes containing *Legionella* undergo the endocytic maturation pathway by default and avoid subsequent degradation by lysosomes [8]. Instead, LCVs extensively communicate with vesicles from the endocytic, secretory, and retrograde trafficking pathways [8]. The LCV eventually develops into a compartment resembling the endoplasmic reticulum (ER) that supports bacterial replication [8]. A specialized type IV secretion system termed Dot/Icm (<u>d</u>efect in <u>o</u>rganelle <u>t</u>rafficking/<u>i</u>ntra<u>c</u>ellular <u>m</u>ultiplication) is highly conserved among *Legionella* species and is critical for LCV biogenesis and intracellular bacterial growth [9]. This apparatus comprises 27 proteins and injects a set of effector proteins across both bacterial and vacuole membranes into the cytosol of infected host cells or the vacuole lumen [10]. These effectors play important roles in the hijacking of a variety of host signaling pathways to create a safe intracellular environment for proliferation [9]. Pathogenesis studies of *Legionella* infections are primarily based on *L. pneumophila*. To date, over 330 Dot/Icm substrates have been identified in *L. pneumophila*. Functional studies of these effector proteins have provided important mechanistic insights into the virulence strategies employed by *L. pneumophila* [9]. For example, many effectors function to modulate host vesicle trafficking between the ER and Golgi apparatus, especially through the manipulation of Rab1 activity [9]. Effector SidM/DrrA is a guanine nucleotide exchange factor (GEF) that promotes the recruitment of Rab1 to the LCV membrane [11, 12], whereas effector LepB acts as a GTPase-activating protein (GAP) to disassociate Rab1 from the LCV [13]. In addition, Rab1 is also subjected to several reversible posttranslational modifications conducted by effector protein pairs DrrA/SidD [14–16], AnkX/Lem3 [17, 18], and SidEs/Dups [19–22], which coordinately impact Rab1 dynamics on the LCV.

Unlike that in *L. pneumophila*, the molecular pathogenesis of nonpneumophila strains is not well studied. Several recent and extensive genomic analyses have revealed profound genetic variations among *Legionella* species [23–25]. Although the Dot/Icm secretion system is present in all sequenced *Legionella* strains, the effector repertoire differs surprisingly. Among the 18,000 Dot/Icm substrates identified in the *Legionella* genus genome, only 8 effectors were found to be shared by each species [25]. Over 66% of reported *L. pneumophila* Dot/Icm effectors are absent in *L. longbeachae*, while 51 novel effectors have been identified [4]. For instance, *L. longbeachae* lacks the Rab1 regulators SidM/DrrA, AnkX, SidD, and Lem3, but SidEs and LepB are present [4, 23]. To date, little is known about the roles of effector proteins

in *L. longbeachae* biology and infection processes. The only example is the effector protein SidC, which functions similarly to its homolog in *L. pneumophila* to promote the interaction of the LCV with the ER [26]. Despite the translocation of a distinct set of effectors, a recent study illustrated that both *L. pneumophila* and *L. longbeachae* develop similar replicative vacuoles in infected cells, albeit through different mechanisms [27].

Autophagy is a cellular process that occurs in eukaryotic cells and targets cytosolic proteins, lipids and organelles to lysosomes for degradation. During the process, sequestered cargoes are enclosed by a membrane-bound compartment termed the autophagosome [28]. Following fusion with a lysosome, the cargoes are consumed in the autolysosome [28]. Autophagy can be a nonselective process that degrades cytoplasmic components to offset the effects of starvation [29]. In contrast, selective autophagy targets specific substrates for degradation and is divided into different types according to the sequestered cargo (e.g. mitophagy, pexophagy, reticulophagy and xenophagy) [29]. Xenophagy is a host autophagic response for the degradation of invading microbes and is recognized as a conserved host immune response against intracellular pathogens [30]. Moreover, many successful intracellular bacteria have evolved distinct mechanisms to avoid clearance by xenophagy [31]. For example, *L. pneumophila* utilizes the effector protein RavZ, a cysteine protease, to inhibit autophagy through the irreversible deconjugation of LC3 from phosphatidylethanolamine (PE) [32]. To date, the host autophagic response to *L. longbeachae* infection has not been reported. In this study, we found that the RavZ ortholog in *L. longbeachae* RavZ$_{LLO}$ plays important roles in the inhibition of the host autophagy pathway. In addition, RavZ$_{LLO}$ decreases cellular polyubiquitin levels when ectopically expressed in mammalian cells and regulates the association of polyubiquitinated species with the LCV in *L. longbeachae*-infected cells.

## Materials and methods

### Bacterial strains, plasmids, and growth conditions

The strains, plasmids, and primers used in this study are provided in S1–S3 Tables, respectively. *Escherichia coli* strains were grown in Luria-Bertani (LB) medium or LB agar plates. When needed, ampicillin (100 μg/mL), kanamycin (30 μg/mL), or chloramphenicol (30 μg/mL) was added to the *E. coli* cultures. *L. pneumophila* (Lp02), *L. longbeachaea* ATCC33462, and their derivatives were cultured on charcoal yeast extract (CYE) agar plates or in N-(2-acetamido)-2-aminoethanesulfonic acid-buffered yeast extract broth (AYE) at 37˚C. If needed, antibiotics were supplemented to the *Legionella* cultures at the following concentrations: ampicillin (100 μg/mL), kanamycin (20 μg/mL), streptomycin (50 μg/mL), and chloramphenicol (5 μg/mL). When necessary, thymidine was added at 200 μg/mL. The in-frame deletion mutants *L. pneumophila* Δ*ravZ*, *L. longbeachae* Δ*ravZ$_{LLO}$*, and *L. longbeachae* Δ*dotB* were generated by allelic exchange as described previously [33, 34]. To determine the translocation of RavZ$_{LLO}$, fragments amplified from *L. longbeachae* genomic DNA were inserted into pXDC61m [35], and the resulting plasmid was introduced into relevant *Legionella* strains by electroporation (2.5 kV; 200 Ω; 0.25 μF). pXDC61JQ was modified based on the backbone of pDXC61m and used for IPTG-inducible expression of Flag-tagged proteins in *L. longbeachae*. To achieve this goal, we inserted a Nde-Flag-BamH-Bgl-Sac-Xho-Sal-Hind polylinker into Nde/HindIII-digested pDXC61m to remove the TEM gene. The *ravZ$_{LLO}$* gene was then cloned into pXDC61JQ and electroporated into *L. longbeachae* strains as described above. For complementation of *L. pneumophila* strains, *ravZ$_{LP}$* and *ravZ$_{LLO}$* were inserted into pZL507 [36] and similarly transformed into *L. pneumophila*. For the expression of proteins in mammalian cells, amplified DNA products were cloned into pCMV-4×Flag [37] or peGFPC1 by standard cloning methods. For protein expression in *E. coli*, genes were inserted into pET28a.

Site-directed mutagenesis of *ravZ_{LLO}* was performed by the Quikchange kit (Agilent) with primers designed by the QuikChange Primer Design program ([https://www.agilent.com.cn/store/primerDesignProgram.jsp](https://www.agilent.com.cn/store/primerDesignProgram.jsp)). The integrity of all plasmids was confirmed by DNA sequencing.

## Recombinant protein purification and *in vitro* deubiquitination assay

*E. coli* BL21 (DE3) harboring pET28-RavZ was cultured to an $OD_{600}$ of 0.6–0.8 at 37˚C in LB broth. After isopropyl β-D-1-thiogalactopyranoside (IPTG) was added at a final concentration of 0.5 mM, the culture was further incubated in a shaker (220 rpm/min) at 16˚C for 16–18 h. The bacterial cells were collected by centrifugation and resuspended in lysis buffer (50 mM $NaH_2PO_4$, 300 mM NaCl, 10 mM imidazole, pH 8.0). Cells were lysed twice using a JN-Mini Low Temperature Ultrahigh Pressure Continuous Flow Cell Cracker (JNBIO, Guangzhou, China). Cell lysates were cleared by centrifugation at 12000×*g* for 20 min. The supernatant containing the protein of interest was mixed with $Ni^{2+}$-NTA beads and incubated at 4˚C for 1 h with end-to-end rotation. The beads were washed with 10 bead volumes of washing buffer (50 mM $NaH_2PO_4$, 300 mM NaCl, 20 mM imidazole, pH 8.0), and 6×His-tagged proteins were eluted with elution buffer (50 mM $NaH_2PO_4$, 300 mM NaCl, 250 mM imidazole, pH 8.0). Eluted proteins were dialyzed twice to remove imidazole in a buffer containing 25 mM Tris-HCl (pH 7.5), 150 mM NaCl and 10% (v/v) glycerol. Protein concentration was measured by the Bradford assay using BSA levels for normalization.

During the diubiquitin cleavage assay, 1 μM purified $His_6$-$RavZ_{LLO}$ was incubated with each diubiquitin (1 μM, Boston Biochem) in 20 μL DUB buffer (50 mM Tris-HCl (pH 7.5), 100 mM NaCl, and 1 mM DTT). The reaction was allowed to proceed for 2 h at 37˚C and was stopped by the addition of 5×SDS loading buffer. The cleavage of diubiquitin was analyzed by Coomassie brilliant blue staining.

## Cell culture, transfection, and immunoprecipitation

HEK293T, HEK293, human lung epithelial cells (A549) and HeLa cells were cultured in Dulbecco's modified minimum Eagle's medium (DMEM) supplemented with 10% fetal bovine serum (FBS). Transfection was conducted using Lipofectamine 3000 (Life Technology) when the cells grew to approximately 80% confluence following the manufacturer's instructions. RAW264.7 and U937 cells were maintained in RPMI 1640 medium containing 10% FBS. At 36 h prior to bacterial infection, U937 cells were differentiated with phorbol-12-myristate-13-acetate (PMA) as described earlier.

To determine the deubiquitination or deneddylation activity of RavZ in cells, HA-Ub or Flag-Nedd8 was cotransfected with GFP-$RavZ_{LP}$, GFP-$RavZ_{LLO}$ or GFP-$RavZ_{LLOC251A}$ into HEK293T cells. GFP-$SdeA_{Dub}$ and GFP-SENP8 were used as positive controls for deubiquitination and deneddylation, respectively. Twenty-four hours after transfection, transfected cells were harvested and lysed in NP40 lysis buffer for 10 min on ice. Then, the lysates were centrifuged at 12,000×*g* at 4˚C for 10 min to remove insoluble fractions. Polyubiquitinated and polyneddylated species were enriched by anti-HA and anti-Flag agarose beads at 4˚C for 8 h on a rotatory shaker, respectively. After washing 3 times with lysis buffer, the beads were resuspended in 50 μL of 1×SDS loading buffer and boiled for 5 min at 100˚C.

To detect the cellular localization of $RavZ_{LLO}$, HeLa cells were transfected with GFP-$RavZ_{LLO}$ for 24 h. Samples fixed with 4% paraformaldehyde were immunostained with organelle-specific primary antibodies. To investigate the influence of RavZ on LC3 puncta formation under transfection conditions, HEK293 cells were cotransfected with GFP-LC3 and 4×Flag-tagged $RavZ_{LP}$, $RavZ_{LLO}$ or $RavZ_{LLOC/A}$ for 24 h. After the cells were fixed with 4%

paraformaldehyde, the percentage of LC3 puncta was measured under a fluorescence microscope.

## Bacterial infections

During infection experiments, overnight liquid cultures of stationary-phase *Legionella* bacteria ($OD_{600}$ = 3.3–3.8) were used to infect cells at the indicated MOIs. To evaluate the translocation of $RavZ_{LLO}$, RAW264.7 cells seeded in 96-well plates at a density of $1\times10^6$ cells/well were infected with relevant *Legionella* strains expressing TEM-$RavZ_{LLO}$ at an MOI of 50. At 2 h post-infection, cells washed with Hanks Balanced Salt Solution (HBSS) were treated with HBSS containing 20 μL of 6×CCF4/AM solution (LiveBLAzer-FRET B/G Loading Kit, Invitrogen). After incubation in the dark for 2 h, translocation of the TEM fusion protein was inspected under a fluorescence microscope (IX83, Olympus) equipped with a β-lactamase FL-Cube (U-N41031, Chroma Technology Corp, Bellows Falls, VT).

To assess the intracellular replication of *L. longbeachae*, U937-derived macrophage cells seeded in 24-well plates were challenged with WT *L. longbeachae*, Δ*dotB*, and Δ*ravZ_{LLO}* at an MOI of 10 in triplicate. At 2 h post-incubation, extracellular bacteria were removed by washing the samples with warm PBS three times. After fresh culture medium was added, the infections were allowed to proceed for 24, 48, and 72 h. Cells were collected at each assay time point and lysed with 0.2% saponin. Serial dilutions of the lysates were plated on CYE plates and grown for 4 days at 37˚C before the CFUs were enumerated.

To inspect LCV-associated $RavZ_{LLO}$ or polyubiquitinated species, $4\times10^5$ PMA-differentiated U937 cells seeded on glass coverslips in 24-well plates were infected with relevant *Legionella* strains at an MOI of 10 for 2 h. After they were extensive washed with PBS, samples fixed with 4% paraformaldehyde were immunostained using the indicated antibodies specific for *L. longbeachae*, Flag, or polyubiquitin.

To assess the impact of RavZ on LC3-II levels in infected cells, HEK293 cells transfected to express the FcγII receptor were infected with the indicated opsonized *Legionella* strains at an MOI of 50 for 2 h in the presence of 160 nM bafilomycin A1 (Sigma) as described earlier. Lysates were prepared by lysing the cells with NP40 lysis buffer, and the level of LC3-II was measured by immunoblotting analysis. To assess the formation of LC3 puncta in infected cells, A549 cells were transfected with mCherry-LC3 and FcγII receptor. At 24 h post-transfection, anti-*Legionella* antibody opsonized bacteria were added to the cells supplemented with 160 nM bafilomycin A1. After 2 h of infection, samples fixed with 4% paraformaldehyde were immunostained with the appropriate antibodies.

## Antibodies, Western blot analysis and immunostaining

For Western blot analysis, proteins were separated by SDS–PAGE and transferred to nitrocellulose membranes (Pall Life Sciences). The membrane was blocked with 5% nonfat milk in PBS for 1 h at room temperature prior to incubation with primary antibodies. Antibodies recognizing $RavZ_{LLO}$ were produced by the immunization of rabbits with purified $His_6$-$RavZ_{LLO}$ following a standard protocol and diluted at 1:2,000 (AbMax Biotechnology Co., LTD, Beijing, China). The source and dilutions for other primary antibodies used in this study were as follows: anti-TEM1 (Abcam, ab12251, 1:3,000), anti-Flag (Sigma, F1804, 1:3,000), anti-LC3 (Sigma, SAB5701328, 1:3,000), anti-GFP (Sigma, G7781, 1:5,000), anti-HA (Santa Cruz, sc-7392, 1:1,000), anti-His (Proteintech, 66005-1-Ig, 1:10,000), anti-ICDH (1:20,000) [33], and anti-tubulin (DSHB, E7, 1:10,000). After the primary antibodies were removed, the membrane was further probed with appropriate fluorescence dye-conjugated secondary antibodies (Li-

Cor). The Odyssey CLx system (Li-Cor) was used to detect and acquire the protein band signals.

During the immunostaining experiments, the cells were fixed with 4% paraformaldehyde in PBS for 30 min at room temperature (RT). After permeabilization with 0.2% Triton X-100 for 5 min, the samples were stained with the appropriate antibodies for 1 h at RT. Rat and rabbit anti-*L. longbeachae* antibodies were commercially generated by AbMax Biotechnology Co. (Beijing, China) and diluted at 1:2,000 for immunostaining. Other primary antibodies used in this study were as follows: anti-*L. pneumophila* (1:10,000), anti-Flag (Sigma, cat# F1804, 1:200), anti-FK1 (Enzo, cat# BML-PW8805, 1:1,000), and anti-calnexin (Abcam, ab22595, 1:100). After the samples were washed with PBS 3 times, they were incubated with appropriate secondary antibodies conjugated to specific fluorescent dyes for 1 h at RT. Fluorescent images were acquired by an Olympus IX-83 fluorescence microscope.

### Bioinformatics

RavZ$_{LLO}$ (LLO_2508) was analyzed by the HHpred server using default parameters (https://toolkit.tuebingen.mpg.de/#/tools/hhpred) [38].

### Data analysis

ImageJ was used to quantify protein abundance. Prism 8.0 (GraphPad, USA) with two-tailed Student's *t* test was used to analyze the data. Results with *P* values less than 0.05 were considered significant.

## Results

### RavZ$_{LLO}$ is translocated into host cells by the Dot/Icm secretion system

*L. pneumophila* effector protein RavZ is a cysteine protease that functions to inhibit the host autophagy response by the deconjugation of lipidated LC3 [32]. Genes coding for RavZ are present in 4 out of the 41 sequenced genomes covering 38 *Legionella* species [23]. Although more than 66% of *L. pneumophila* effectors are missing in *L. longbeachae* [4], the RavZ ortholog exists in *L. longbeachae* and is encoded by *llo_2508* (*ravZ$_{LLO}$*). RavZ$_{LLO}$ shares 59.4% overall sequence identity with RavZ from *L. pneumophila*, and the catalytic cysteine residue for the enzymatic activity is conserved between the two proteins (S1 Fig). To determine whether RavZ$_{LLO}$ is a bona fide Dot/Icm substrate of *L. longbeachae*, we employed the β-lactamase reporter assay that has been successfully applied to identify and validate effector proteins of *Legionella* spp. [27, 35, 39]. Infection of RAW264.7 cells with *L. longbeachae* producing TEM--RavZ$_{LLO}$ led to obvious translocation of the fusion protein, which was evidenced by 27% of the cells showing blue fluorescence signals emitted by the β-lactamase substrate CCF4-AM (Fig 1A). In addition, samples infected by the wild-type *L. pneumophila* strain harboring TEM--RavZ$_{LLO}$ displayed blue cells at a proportion of 35% (Fig 1A). In contrast, despite producing considerable levels of TEM-RavZ$_{LLO}$, translocation of the fusion protein did not occur when cells were challenged with the *dot/icm*-deficient strains (Fig 1A and 1B). Taken together, our data demonstrated that RavZ$_{LLO}$ is an effector protein recognized by the Dot/Icm machinery of both *L. longbeachae* and *L. pneumophila*.

### RavZ$_{LLO}$ is associated with the LCV and is dispensable for the intracellular growth of *L. longbeachae*

To test whether RavZ$_{LLO}$ is targeted to any specific organelles in human cells, we generated green fluorescence protein (GFP)-tagged RavZ$_{LLO}$ and ectopically expressed the construct in

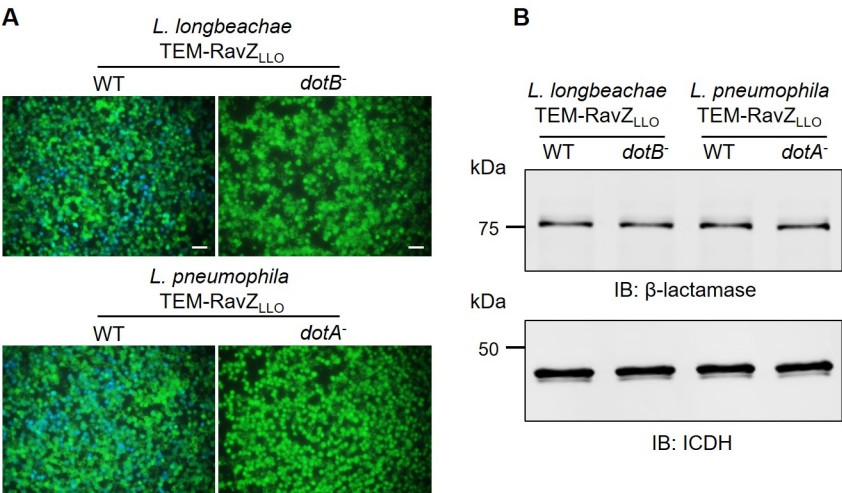

**Fig 1. Dot/Icm-dependent translocation of RavZ$_{LLO}$ into host cells.** (A) RAW264.7 cells were infected with WT or Dot/Icm-deficient *Legionella* strains expressing TEM-RavZ$_{LLO}$ for 2 h, and the translocation of proteins was detected by CCF4/AM dye and visualized by a fluorescence microscope. Bar, 50 μm. (B) Expression of TEM-RavZ$_{LLO}$ was confirmed by probing bacterial lysates with antibodies specific for β-lactamase. Isocitrate dehydrogenase (ICDH) was used as a loading control. Data shown in panels (A) and (B) are representative of three different experiments.

HeLa cells by transfection. The subcellular localization of RavZ$_{LLO}$ was analyzed under fluorescence microscopy after cells were immunostained with antibodies specific for different organelle markers. GFP-RavZ$_{LLO}$ expression showed extensive overlap with the ER-resident protein calnexin, suggesting that RavZ$_{LLO}$ is targeted to the ER (Fig 2A).

Next, we investigated the cellular distribution of RavZ$_{LLO}$ during infection. We generated a plasmid bearing Flag-tagged RavZ$_{LLO}$ and transformed it into both WT and *dotB$^-$* mutant *L. longbeachae* strains. U937 cells were then infected with these strains and immunostained with Flag antibodies. After 2 h of infection, Flag-RavZ$_{LLO}$ staining signals were detected in 84% of the LCVs harboring WT *L. longbeachae* (Fig 2B–2D). In contrast, despite similar Flag-RavZ$_{LLO}$ expression levels, few LCVs were positively stained with Flag antibody when the cells were infected with the *L. longbeachae* strain lacking a functional Dot/Icm system (Fig 2B–2D). These data indicate that RavZ$_{LLO}$ is associated with the LCV after being translocated into infected cells by *L. longbeachae*.

To characterize the role of RavZ$_{LLO}$ in bacterial infection, we first made a *ravZ$_{LLO}$* deletion mutant of the *L. longbeachae* strain by allelic exchange. Then, the intracellular replication of the mutant strain was monitored in U937 cells. After 3 days of infection, approximately 3 orders of magnitude more WT *L. longbeachae* were recovered, whereas strains with the absence of *dotB* failed to replicate in the host cells (S2 Fig). Notably, the deletion of *ravZ$_{LLO}$* in *L. longbeachae* did not cause a significant growth defect at any assayed time point (S2 Fig). These results suggest that *ravZ$_{LLO}$* is dispensable for the intracellular proliferation of *L. longbeachae*.

## Ectopic expression of RavZ$_{LLO}$ in mammalian cells inhibits host autophagy

Considering that RavZ inhibits the host autophagy response and the high sequence similarity between RavZ$_{LP}$ and RavZ$_{LLO}$, we reasoned that RavZ$_{LLO}$ and RavZ$_{LP}$ may exhibit identical biological functions. To this end, we cotransfected HeLa cells with GFP-LC3 and Flag-tagged RavZ, and the formation of LC3 puncta was measured under a fluorescence microscope. In the

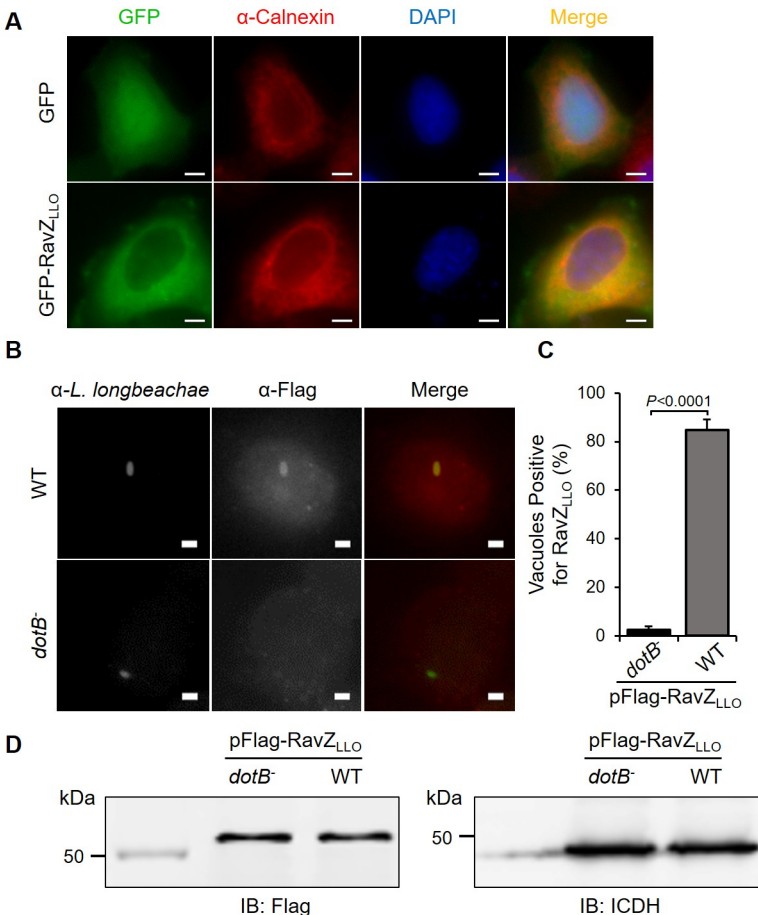

**Fig 2. Cellular localization of RavZ_{LLO}.** (A) Ectopically expressed RavZ_{LLO} localized to the ER. HeLa cells transfected to express GFP-RavZ_{LLO} were fixed and immunostained with antibodies specific for calnexin (red). The nuclei were stained with DAPI (4', 6'-diamino-2-phenylindole) (dark blue). Fluorescent signals were inspected by an Olympus IX-83 fluorescence microscope. Bar, 5 μm. (B) Representative LCVs with positive-association of RavZ_{LLO} during *L. longbeachae* infection. RavZ_{LLO} is associated with the LCV during *L. longbeachae* infection. U937 cells were challenged with *L. longbeachae* strains expressing Flag-RavZ_{LLO} for 2 h. Following sequential immunostaining of the fixed samples with antibodies specific for *L. longbeachae* and Flag, fluorescent images were taken by a fluorescence microscope. Bar, 2 μm. (C) Percentages of Flag-RavZ_{LLO}-positive LCVs shown in (B). At least 100 vacuoles were quantified for each infection sample. (D) Expression of Flag-RavZ_{LLO} in *L. longbeachae* strains was detected by Western blot with the Flag-specific primary antibody. The ICDH was probed as a loading control. (A), (B) and (D) are representative images from three independent experiments. Data shown in (C) are the mean ± standard deviation (SD) and are one representative of three independent experiments performed in triplicate. The *P* values were calculated by the two-tailed Student's *t* test, and *P* values of <0.05 represent a significant difference.

absence of RavZ, 75% of the cells showed punctate LC3-positive autophagosomes (APs), which was a result of basal levels of autophagy (Fig 3A–3C). In contrast, the percentage of cells with LC3 puncta was reduced to 9% when RavZ_{LP} was coexpressed (Fig 3A–3C). Similarly, the number of LC3 puncta-containing cells was significantly decreased in the presence of RavZ_{LLO} (Fig 3A–3C). Additionally, mutation of Cys251 to Ala in RavZ_{LLO} remarkably alleviated its ability to inhibit host autophagy (Fig 3A–3C).

RavZ behaves as a cysteine protease that irreversibly uncouples PE-conjugated LC3 on AP membranes. Therefore, we further evaluated RavZ_{LLO}-mediated autophagy evasion by measuring the levels of the lipidated form of LC3 (LC3-II) by Western blot analysis. Notably, although less prominent than RavZ_{LP} expression, the expression of GFP-RavZ_{LLO} indeed

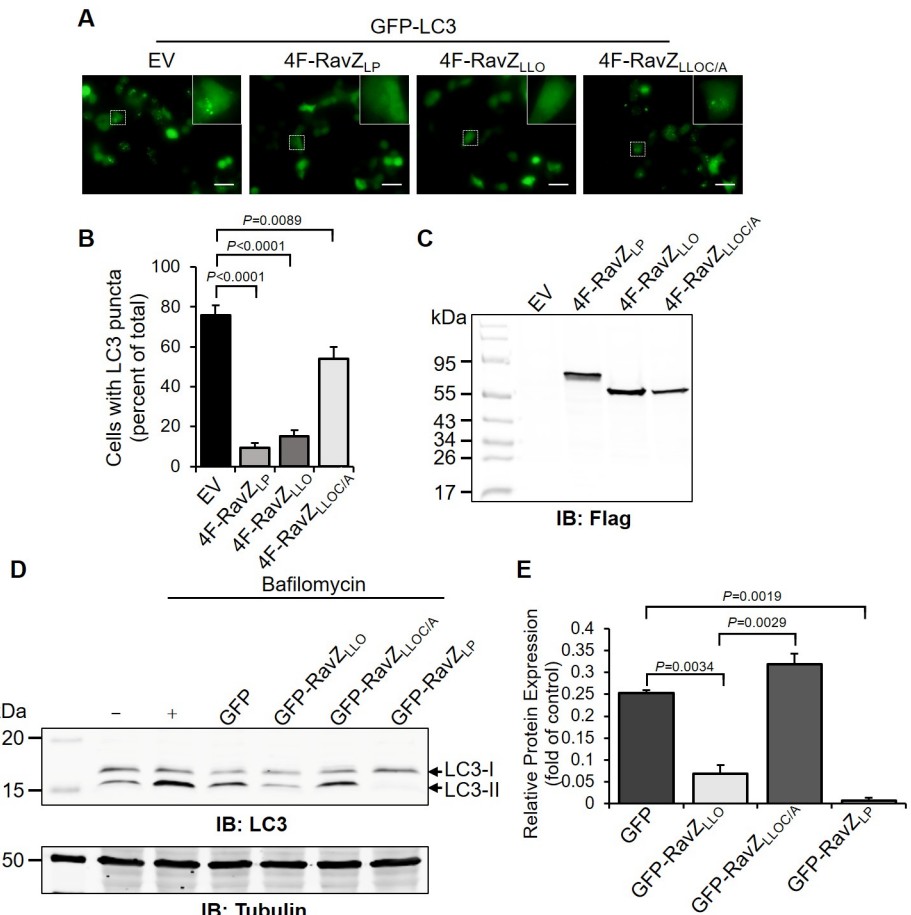

**Fig 3. Ectopic expression of RavZ$_{LLO}$ in mammalian cells suppresses host autophagy.** (A) Fluorescent images of LC3 puncta formation in HeLa cells cotransfected with plasmids expressing GFP-LC3 and 4F-RavZ$_{LP}$, 4F-RavZ$_{LLO}$, or 4F-RavZ$_{LLOC/A}$. Samples were fixed at 24 h post-transfection, and LC3 puncta were visualized under a fluorescence microscope. Insets represent 10× magnification of the dashed-lined regions. EV indicates empty vector. Scale Bar, 20 μm. (B) Percentage of cells with LC3 puncta as shown in (A); at least 100 cells were calculated for each sample. (C) Expression of Flag-tagged protein was confirmed by Western blot with antibodies specific for Flag. (D) Western blot analysis of LC3-I and LC3-II levels in HEK293T cells expressing the indicated GFP fusion proteins. Tubulin was used as a loading control. Cells treated with bafilomycin A1 are indicated. (E) The LC3-II/LC3-I ratios in transfected cells receiving bafilomycin A1 were analyzed by ImageJ. Data shown in (A), (C) and (D) are one representative from three independent experiments. Data shown in (B) and (E) are the mean ± SD of three independent assays. The $P$ values were calculated by two-tailed Student's $t$ test, and $P$ values of <0.05 indicated a significant difference.

significantly reduced endogenous LC3-II levels in cells treated with bafilomycin A1 (Fig 3D and 3E). The cysteine protease activity of RavZ$_{LLO}$ was responsible this decrease since the expression of the catalytically inactive mutant resulted in a considerable level of LC3-II as the GFP control (Fig 3D and 3E).

## RavZ$_{LLO}$ inhibits autophagy in *Legionella* infection

Having proven the functional role of RavZ$_{LLO}$ in autophagy inhibition in transfected cell lines, we further investigated its biological relevance in *L. longbeachae* infection. To this end, we infected HEK293 cells with *L. longbeachae* strains and monitored the LC3-II levels. Similar to *L. pneumophila*, samples infected with the WT *L. longbeachae* strain showed a significant blockage of LC3-II generation (Fig 4A and 4B). However, the LC3-II level was not affected

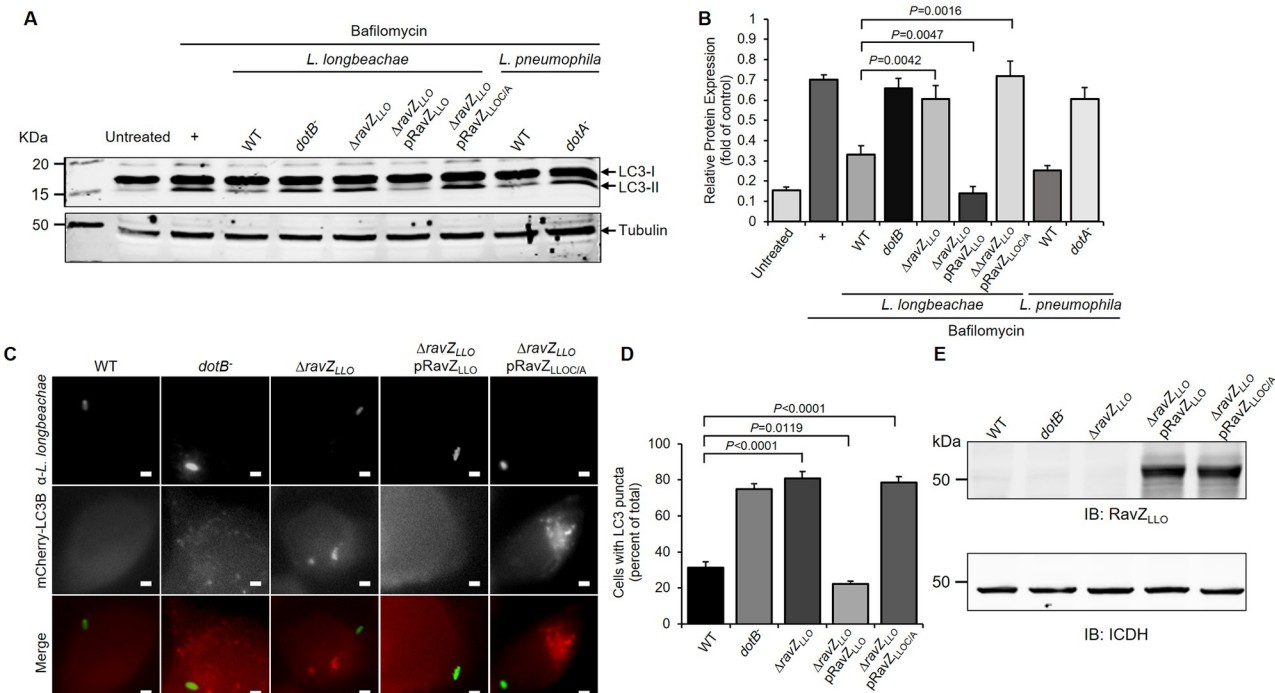

**Fig 4. Suppression of host autophagic response by RavZ_LLO during *L. longbeachae* infection.** (A) HEK293 cells were transfected to express the FcγII receptor and infected with the indicated opsonized *L. longbeachae* strains for 2 h in the presence of bafilomycin A1. The LC3 levels of the infection samples were determined by Western blot with the LC3 antibody. Infection of cells with *L. pneumophila* was included as a control. (B) The LC3-II/LC3-I ratio for each treatment shown in (A) was measured by ImageJ. (C) Representative images of LC3 distribution in mCherry-LC3-expressing A549 cells that were infected with the relevant *L. longbeachae* strains for 2 h. Bar, 2 μm. (D) Percentage of *L. longbeachae*-infected cells containing LC3 puncta. At least 100 cells were scored for each sample. (E) The expression of RavZ_LLO in *L. longbeachae* strains was determined by Western blot with the RavZ_LLO-specific antibody. ICDH was probed as a loading control. The results shown in (A), (C) and (E) are representative of three independent experiments. The results shown in (B) and (D) are the mean ± SD of three independent assays. The *P* values were calculated by two-tailed Student's *t* test, and *P* values of <0.05 indicated a significant difference.

when the cells were challenged with the *dotB* deletion mutant, suggesting Dot/Icm-dependent inhibition of the host autophagy system by *L. longbeachae* (Fig 4A and 4B). Consistent with this observation, infection of A549 cells stably expressing mCherry-LC3 with virulent *L. longbeachae* led to a considerably lower percentage of cells containing punctate LC3-positive APs than infection with the Dot/Icm-deficient mutant (Fig 4C and 4D). The autophagy inhibition that is mediated by *L. longbeachae* could be attributed to the activity of RavZ_LLO because a similar level of LC3-II and percentage of punctate LC3-positive APs in infected cells were detected in samples receiving *L. longbeachae* Δ*dotB* and Δ*ravZ_LLO* (Fig 4A–4E). Autophagy inhibition was completely restored by the introduction of a plasmid encoding *ravZ_LLO* into the Δ*ravZ_LLO* mutant (Fig 4A–4E). However, complement of the mutant strain with catalytically inactive RavZ_LLO failed to suppress the autophagic response in either assay (Fig 4A–4E). Hence, RavZ_LLO is necessary to block autophagy during *L. longbeachae* infection.

Since RavZ_LLO was also recognized by the *L. pneumophila* Dot/Icm system, we analyzed whether the defect in autophagy inhibition displayed by *L. pneumophila* Δ*ravZ* could be complemented by RavZ_LLO. Consistent with previous observations [32], the Δ*ravZ* mutant and Δ*dotA* mutant of *L. pneumophila* were equally defective in autophagy inhibition, which was determined by the levels of lipidated LC3 as well as the proportion of LC3 puncta-positive cells (Fig 5A–5E). Interestingly, both phenotypes were fully restored by the production of not only RavZ_LP but also RavZ_LLO from plasmid-borne genes (Fig 5A–5E). Taken together, our data

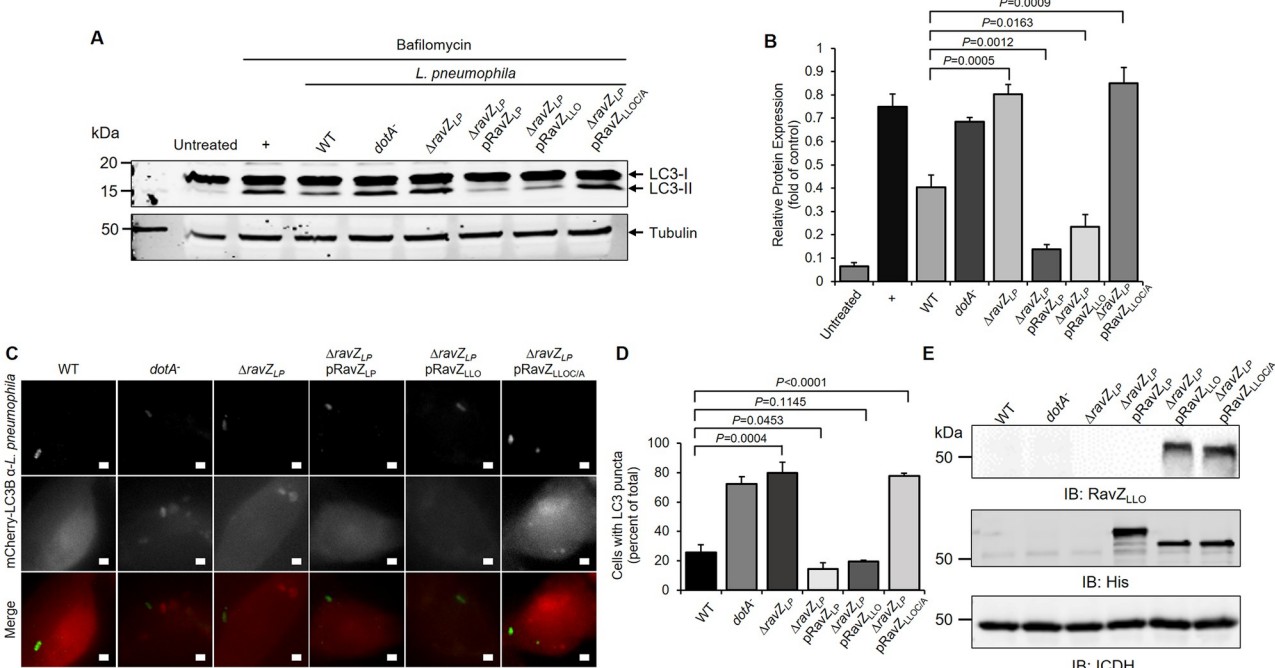

**Fig 5. Expression of RavZ$_{LLO}$ in *L. pneumophila* Δ*ravZ* restores autophagy evasion.** (A) Western blot analysis of the LC3-I and LC3-II levels in HEK293 cells infected for 2 h with the indicated *L. pneumophila* strains. Tubulin was used as a loading control. (B) The LC3-II/LC3-I ratios of the infection samples shown in (A) were quantified by ImageJ. (C) Representative fluorescent images showing LC3 puncta in mCherry-LC3-expressing A549 cells challenged with relevant *L. pneumophila* strains. Bar, 2 μm. (D) Percentage of LC3 puncta-positive cells shown in (C). For each infection sample, at least 100 cells were measured. (E) The production of RavZ$_{LLO}$ in *L. pneumophila* Δ*ravZ* complemented with plasmids coding for WT *ravZ$_{LLO}$* or *ravZ$_{LLOC/A}$* was detected by Western blot with the RavZ$_{LLO}$-specific antibody. The expression of RavZ$_{Lp}$ in *L. pneumophila* Δ*ravZ* strains was determined by antibodies specific for His. ICDH was detected as a loading control. Data shown in (A), (C) and (E) are one representative from three independent experiments. Data shown in (B) and (D) are the mean ± SD of three independent assays. The *P* values were calculated by two-tailed Student's *t* test, and *P* values of <0.05 indicated a significant difference.

indicate that *L. longbeachae* RavZ suppresses the host autophagic response during bacterial infection, while RavZ$_{LP}$ and RavZ$_{LLO}$ are functionally redundant.

## Expression of RavZ$_{LLO}$ in mammalian cells decreases cellular polyubiquitination and polyneddylation

Previous structural studies showed that the N-terminal domain of RavZ harbors a fold similar to that of the Ubl-specific protease (Ulp) family deubiquitinase (DUB) [40]. In addition, bioinformatics analysis of both RavZ$_{LP}$ and RavZ$_{LLO}$ using HHpred revealed distant homology of these proteins with known bacterial DUBs, including SseL from *Salmonella enterica* serovar Typhimurium (S4 Table). The His-Asp-Cys catalytic triad critical for SseL function is conserved in RavZ$_{LLO}$ (S3 Fig). These observations together illustrate the potential role of RavZ$_{LLO}$ in hydrolyzing polyubiquitin chains. To test this hypothesis, we cotransfected HEK293T cells with GFP-RavZ and human influenza hemagglutinin (HA)-tagged ubiquitin and assayed the polyubiquitinated protein levels by immunoblotting. *In vivo* production of RavZ$_{LLO}$ but not RavZ$_{LP}$ significantly reduced the levels of cellular polyubiquitinated species, which was shown by the decreased expression of the high-molecular-weight ubiquitin ladder detected by the anti-HA antibody (Fig 6A). Surprisingly, the substitution of Cys251 to Ala led to a similar decrease in polyubiquitin signals as that observed in the WT RavZ$_{LLO}$ (Fig 6A). Similarly, ectopic expression of RavZ$_{LLO}$ but not RavZ$_{LP}$ decreased cellular polyneddylation, a ubiquitin-

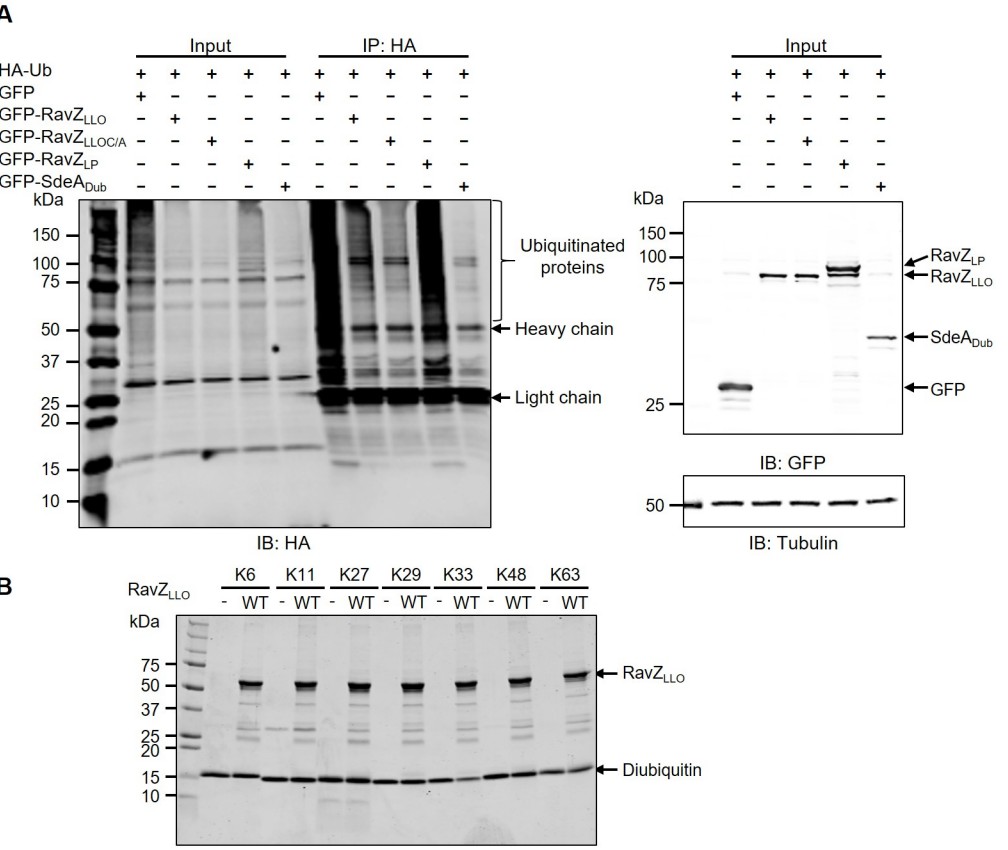

**Fig 6. Expression of RavZ$_{LLO}$ in mammalian cells reduces cellular polyubiquitin levels.** (A) HEK293T cells were cotransfected with HA-Ub and GFP-RavZ$_{LLO}$, GFP-RavZ$_{LLOC/A}$, or GFP-RavZ$_{LP}$, and cellular polyubiquitinated proteins were then immunoprecipitated by anti-HA agarose and measured by Western blotting with an HA antibody. Expression of the GFP fusions was confirmed by probing the cell lysates with the GFP antibody. GFP-SdeA was included as a positive DUB control. (B) Recombinant His$_6$-RavZ$_{LLO}$ purified from *E. coli* was unable to cleave diubiquitins in *in vitro* reactions. His$_6$-RavZ$_{LLO}$ was incubated with the indicated diubiquitins at 37°C for 2 h. Hydrolysis of the substrates was detected by Coomassie brilliant blue staining. Data shown in (A) and (B) are one representative from three independent experiments.

like modification, in a Cys251-independent manner (S4 Fig). These data indicate that RavZ$_{LLO}$ can cleave polyubiquitin and polynedd8 chains, and the catalytic cysteine responsible for deconjugating LC3-PE is not important for such activity.

Next, we aimed to characterize the DUB activity of RavZ$_{LLO}$ in *in vitro* biochemical reactions using a panel of diubiquitins linked by different lysine residues. However, recombinant RavZ$_{LLO}$ obtained from *E. coli* failed to hydrolyze any of the tested diubiquitins (Fig 6B). Therefore, it is possible that RavZ$_{LLO}$ requires a cofactor to activate its DUB activity, as it is ectopically expressed in mammalian cells. Alternatively, RavZ$_{LLO}$ might activate the DUB activity of host proteins to decrease cellular polyubiquitination and polyneddylation.

## RavZ$_{LLO}$ reduces the association of polyubiquitinated species with the LCV

During *L. pneumophila* infection, the LCVs are known to be decorated with polyubiquitinated protein by a process depending on the Dot/Icm system [41]. Despite the lack of established DUB activity, RavZ could interfere with ubiquitin accumulated on *Salmonella*-containing vacuoles in a *Legionella* and *Salmonella* coinfection system [42]. However, RavZ did not affect

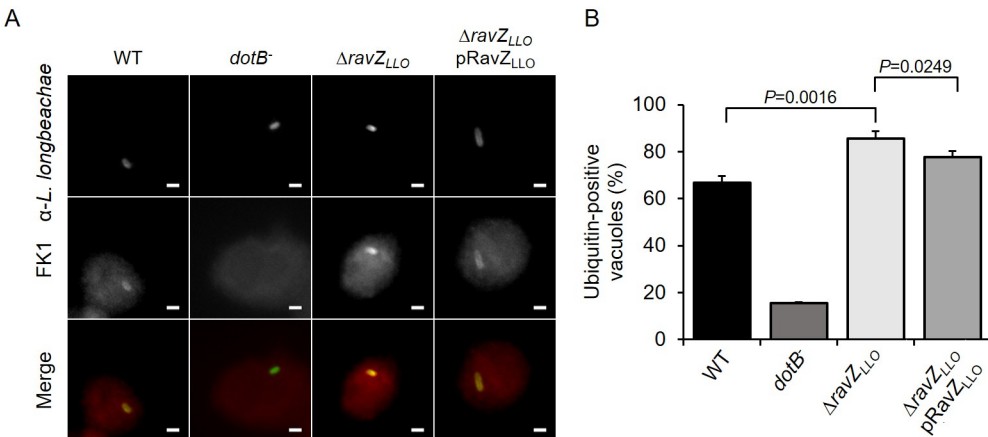

**Fig 7. RavZ$_{LLO}$ decreases the association of polyubiquitinated species with the LCV harboring *L. longbeachae*.** (A) Representative images of polyubiquitinated proteins associated with LCVs. U937 cells infected for 2 h with the indicated *L. longbeachae* strains were sequentially immunostained with antibodies specific for *L. longbeachae* and polyubiquitin (FK1). Fluorescent images were acquired by a fluorescence microscope. Bar, 2 μm. (B) Percentage of polyubiquitin-positive LCVs. At least 100 vacuoles were counted for each of the infection samples. Data shown in (A) are one representative from three independent experiments. Data shown in (B) are the mean ± SD of three independent experiments. The *P* values were calculated by the two-tailed Student's *t* test, and results with *P* values of <0.05 were considered significant.

ubiquitin recruitment to LCVs, which may be attributed to an unknown antagonistic mechanism mediated by *Legionella* [42]. Considering the polyubiquitin deconjugating activity of RavZ$_{LLO}$ in mammalian cells and its localization on bacterial phagosomes during *L. longbeachae* infection, we speculated that this protein might play a role in the regulation of ubiquitin association on the LCV. To test this hypothesis, we infected U937 cells with relevant *L. longbeachae* strains followed by immunostaining with an FK1 antibody that specifically recognizes ubiquitinated proteins. Similar to *L. pneumophila* infection, vacuoles harboring virulent *L. longbeachae* were also enriched with polyubiquitinated species, and 66% of the LCVs were positively stained with FK1 antibodies at 2 h post-infection (Fig 7A and 7B). Importantly, the absence of *ravZ$_{LLO}$* resulted in a significant increase in the proportion of ubiquitin-positive vacuoles (Fig 7A and 7B). Moreover, expression of RavZ$_{LLO}$ from a plasmid in the *L. longbeachae* Δ*ravZ$_{LLO}$* mutant restored ubiquitin association with the LCV (Fig 7A and 7B). Interestingly, although challenging U937 cells with the *L. pneumophila* strain lacking *the ravZ* gene was able to promote the recruitment of ubiquitinated proteins to the bacterial phagosomes as efficiently as challenge with the WT *L. pneumophila*, complement of the *L. pneumophila* Δ*ravZ* mutant with a plasmid-borne *ravZ$_{LLO}$* appeared to considerably decrease the percentage of ubiquitin-positive LCVs (Fig 8A and 8B). These results suggest that RavZ$_{LLO}$ contributes to the removal of ubiquitin from bacterial phagosomes.

## Discussion

The host autophagy pathway not only plays vital roles in maintaining intracellular homeostasis but can also serve as an important cell autonomous immune response that eliminates intracellular bacteria to counteract infection [43]. As a successful intracellular pathogen, *L. pneumophila* has developed various effector-driven mechanisms that may interplay to promote the evasion of bacterial clearance by the host xenophagic response. These strategies include the prevention of autophagosome formation via cleavage of LC3-PE by RavZ [32], disturbance of sphingolipid biosynthesis through the *Legionella* effector sphingosine-1-phosphate lyase

(LpSpl) [44], and degradation of syntaxin 17 via the serine protease activity of effector Lpg1137 [45]. Moreover, a panel of effector proteins could indirectly impact host autophagy through interference with mTORC1 signaling. The SidE effector family and SetA are inhibitors of mTORC1 that can promote the translocation of transcription factor EB (TFEB) into the nucleus [46, 47]. In contrast, the glucosyltransferases in *Legionella* (Lgt1, Lgt2, and Lgt3) activate the mTORC1 pathway by inhibiting translation by targeting host elongation factor 1A [46]. In addition, a recent report illustrated an epigenetic mechanism that suppresses the host autophagic response mediated by the effector Lpg2936, which reduces the expression of autophagy-related genes ATG7 and LC3 during *L. pneumophila* infection [48]. Interestingly, *L. pneumophila* Dot/Icm substrate LegA9 appears to promote recognition of LCV for autophagy uptake and elimination via the induction of ubiquitin accumulation and association of the adaptor protein p62 on the vacuoles [49]. These previous observations revealed the extensive effector cohort and the sophisticated mechanisms that *L. pneumophila* has developed to manipulate host autophagy.

In this study, we demonstrated that, similar to most intracellular pathogens, such as *S.* Typhimurium and *L. pneumophila*, *L. longbeachae* also actively suppresses host autophagy during infection in a Dot/Icm-dependent manner. RavZ$_{LLO}$ in *L. longbeachae* is responsible for autophagy inhibitory activity and may act in the same manner as RavZ$_{LP}$. Indeed, RavZ$_{LLO}$ and RavZ$_{LP}$ are functionally redundant, as the complement of *L. pneumophila* Δ*ravZ* with RavZ$_{LLO}$ efficiently restores its capability to inhibit autophagy. This finding is consistent with an earlier observation that the deficiency in ER recruitment displayed by *L. pneumophila* Δ*sidC/sdcA* can be fully complemented by SidC$_{LLO}$, albeit they share only 40% sequence identity [26]. The deletion of *ravZ$_{LLO}$* did not affect the intracellular proliferation of *L. longbeachae*, suggesting the presence of other effector proteins that promote bacterial evasion of the host autophagy system. Comparative analysis of the effector repertoires revealed that orthologous proteins were found for RavZ, SidEs, Lpg2936, and Lpg1137 but not for Lgts, LpSpl, LegA9, and SetA in *L. longbeachae* (S5 Table), which is indicative of both potential similarities and distinctions between *L. longbeachae* and *L. pneumophila* in the modulation of autophagy.

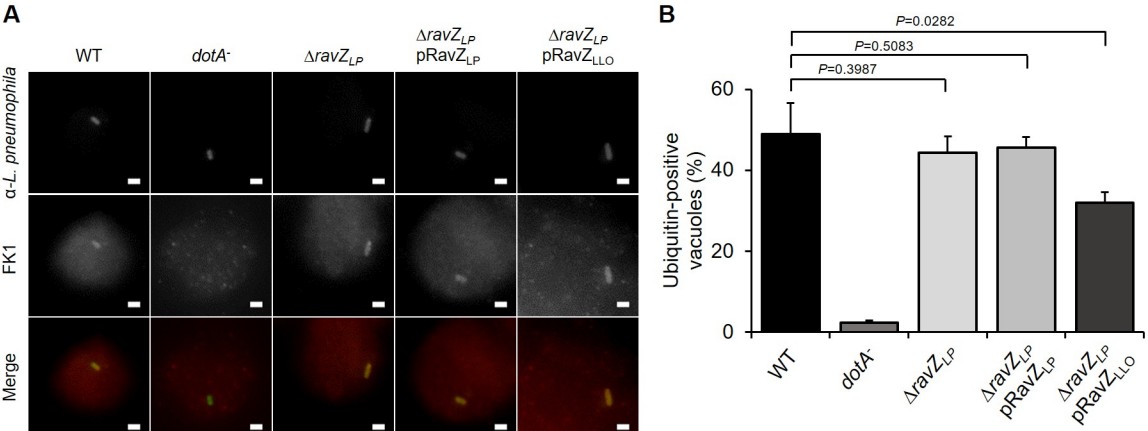

**Fig 8. Complementation of *L. pneumophila* Δ*ravZ* with RavZ$_{LLO}$ affects ubiquitin recruitment to the LCV.** (A) Representative images of polyubiquitinated proteins associated with LCVs. U937 cells were challenged with WT *L. pneumophila*, *dotA$^-$* mutant, Δ*ravZ*, or Δ*ravZ* complemented with RavZ$_{LP}$/RavZ$_{LLO}$ for 2 h. The cells were immunostained with antibodies against *L. pneumophila* and FK1. Images were acquired by a fluorescence microscope. Bar, 2 μm. (B) Quantification of the LCVs that were positively stained with the FK1 antibody. At least 100 vacuoles were scored for each sample. Data shown in (A) are one representative from three independent experiments. Data shown in (B) are the mean ± SD of three independent experiments. The *P* values were calculated by the two-tailed Student's *t* test, and results with *P* values of <0.05 were considered significant.

Previous structural analyses of RavZ uncovered the similarity of its N-terminal domain with that in the Ulp family DUB-like enzymes [40]. Furthermore, coinfection of *L. pneumophila* and *S.* Typhimurium demonstrated the roles of RavZ in preventing ubiquitin recruitment to SCVs [42]. Paradoxically, RavZ cannot reduce the levels of ubiquitin on the LCVs in *L. pneumophila*-infected cells, possibly due to the presence of an antagonistic strategy used in LCVs to protect against RavZ [42]. These observations raise the possibility that, in addition to targeting LC3-PE, RavZ may possess DUB activity to deconjugate certain ubiquitinated substrates on SCVs. However, the DUB activity of RavZ was not established, as ectopic expression of RavZ did not affect cellular polyubiquitin levels, indicating an indirect role of RavZ in ubiquitin removal from SCVs [42]. Intriguingly, although it is not conclusive that RavZ$_{LLO}$ is a canonical DUB, ectopic expression of RavZ$_{LLO}$ in mammalian cells could decrease the levels of polyubiquitinated proteins through a mechanism that is independent of the catalytic cysteine essential for uncoupling LC3-PE. Consistently, RavZ$_{LLO}$ has been shown to regulate the association of polyubiquitinated species with LCVs during *L. longbeachae* and *L. pneumophila* infection. The ubiquitin feature of LCVs creates a signaling platform that promotes the recruitment of autophagy adaptors, including p62/SQSTM1, NDP52, NBR1, and optineurin [50]. Notably, changes in the level or nature of the polyubiquitin on the LCV reasonably affects adaptor recruitment. SidE family effector proteins can exclude adaptor proteins associated with the LCV, possibly by the generation of noncanonical ubiquitin linkages [51]. In contrast, LegA9 targets LCVs for autophagy uptake through increased ubiquitin labeling and p62/SQSTM1 recruitment of the LCV [49]. Hence, in addition to promoting the deconjugation of LC3-PE, RavZ$_{LLO}$ may facilitate autophagy evasion by reducing the levels of ubiquitinated proteins on the bacterial phagosome.

## Supporting information

**S1 Fig. Sequence alignment of RavZ$_{LP}$ and RavZ$_{LLO}$.** Alignment was performed by Clustal Omega (https://www.ebi.ac.uk/Tools/msa/clustalo/) and ESPript 3.0 (https://espript.ibcp.fr/ESPript/cgi-bin/ESPript.cgi). The catalytic residues Cys258 in RavZ$_{LP}$ and Cys251 in RavZ$_{LLO}$ are highlighted by a black box.
(TIF)

**S2 Fig. RavZ$_{LLO}$ is dispensable for the intracellular replication of *L. longbeachae*.** U937 cells seeded in 24-well plates were challenged with WT, Δ*dotB*, and Δ*ravZ$_{LLO}$ L. longbeachae* strains at an MOI of 10. At the indicated timepoints, infected cells were lysed by saponin and plated on CYE plates. The CFUs were counted after growing the bacteria at 37°C for 4 days. The results are from one representative experiment performed in triplicate from three independent experiments. Data are presented as the mean ± SD.
(TIF)

**S3 Fig. Sequence alignment of RavZ$_{LLO}$ with the *S.* Typhimurium effector protein SseL.** The alignment was generated by HHpred, and the catalytic His-Asp-Cys residues are highlighted by red boxes.
(TIF)

**S4 Fig. Expression of RavZ$_{LLO}$ in mammalian cells reduces cellular polyneddylation.** HEK293T cells were transfected to coexpress Flag-Nedd8 and each of the indicated GFP fusion proteins. At 24 h post-transfection, cells were lysed and subjected to immunoprecipitation by anti-Flag agarose. The levels of cellular polyneddylated proteins were measured by Western blot with the Flag antibody. Anti-GFP and anti-tubulin blots were performed to confirm the expression of the GFP fusion proteins and ensure equal loading of proteins, respectively.

GFP-SENP8 was included as the positive polyneddylation control. The data shown are representative of three independent experiments.
(TIF)

**S1 Table. Bacterial strains used in the study.**
(DOCX)

**S2 Table. Plasmids used in the study.**
(DOCX)

**S3 Table. Primers used in the study.**
(DOCX)

**S4 Table. Bioinformatics analysis of RavZ$_{LLO}$ via HHpred.**
(DOCX)

**S5 Table. *L. longbeachae* effector proteins potentially modulate host autophagy pathway.**
(DOCX)

**S1 Raw images.**
(PDF)

## Author Contributions

**Data curation:** Yunjia Shi, Hongtao Liu, Kelong Ma.

**Funding acquisition:** Jiazhang Qiu.

**Methodology:** Zhao-Qing Luo.

**Resources:** Jiazhang Qiu.

**Supervision:** Zhao-Qing Luo, Jiazhang Qiu.

**Validation:** Yunjia Shi.

**Writing – original draft:** Yunjia Shi, Jiazhang Qiu.

**Writing – review & editing:** Yunjia Shi, Hongtao Liu, Zhao-Qing Luo, Jiazhang Qiu.

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
