## [Decision Letter · Decision Letter 0]

12 Dec 2022

PONE-D-22-30067Legionella longbeachae effector protein RavZ inhibits autophagy and regulates phagosome ubiquitination during infectionPLOS ONE

Dear Dr. Qiu,

Thank you for submitting your manuscript to PLOS ONE. After careful consideration, we feel that it has merit but does not fully meet PLOS ONE’s publication criteria as it currently stands. Therefore, we invite you to submit a revised version of the manuscript that addresses the points raised during the review process.

Your manuscript has now been reviewed and the reviewer comments are appended below. You will see that they have raised points that need to be addressed by a minor revision.

We look forward to receiving your revised manuscript.

Kind regards,

Aristóbolo M Silva

Academic Editor

PLOS ONE

Journal Requirements:

"This work was supported by National Natural Science Foundation of China (grant #: 31970134 and 32170182 to Jiazhang Qiu); the Thousand Young Talents Program of the Chinese government (Jiazhang Qiu) and startup fund from Jilin University and the First Hospital of Jilin University."

Reviewers' comments:

Reviewer's Responses to Questions

**Comments to the Author**

1. Is the manuscript technically sound, and do the data support the conclusions?

Reviewer #1: Partly

Reviewer #2: Yes

2. Has the statistical analysis been performed appropriately and rigorously? 

Reviewer #1: Yes

Reviewer #2: Yes

3. Have the authors made all data underlying the findings in their manuscript fully available?

Reviewer #1: Yes

Reviewer #2: Yes

4. Is the manuscript presented in an intelligible fashion and written in standard English?

Reviewer #1: Yes

Reviewer #2: Yes

5. Review Comments to the Author

Reviewer #1: The manuscript describes the functional characterization of RavZ protein from Legionella longbeachae (RavZLLO) and its comparison with the RavZ ortholog from L. pneumophila (RavZLP). The authors demonstrate that RavZLLO is a secreted effector and interfere with host autophagy by removing the lipid anchor from LC3 protein, similarly to RavZLP. Moreover, RavZLLO can substitute for RavZLP in L. pneumophila. Next, the authors show that presence in the human cell RavZLLO but not RavZLP, either through direct expression or by infection with bacteria, decreases the level of polyubiquitination and polyneddylation and hypothesize that RavZLLO has deubiquitination activity. They support this view by noting that the N-terminal domain of RavZ displays fold similarity to Ubl-specific cysteine proteases, including well characterized Salmonella Typhimurium SseL.

While they observe decrease in the polyubiquitinated proteins in the human cell when RavZLLO is present, the same effect is caused by the mutant in which the active site Cys251 is replaced by Ala. In the alignment with SseL, Cys251 corresponds to active site cysteine in SseL. Moreover, no activity on various di-ubiquitins was demonstrated in vitro. The authors suggests that an association with an unknown host protein activated its deubiquitination activity. This is hand waving argument, and a more likely scenario is that RavZLLO stimulated deubiquitinase activity of another protein. That seems to me more reasonable explanation since RavZLP does not show this behavior, has closely similar N-terminal domain and differs in the C-terminal segment.

My other comment is to the discussion section. The first few pages repeat some of the introduction and are not relevant to the presented results, which are only discussed on pp.18-19.

Figures:

Figure 2B – what is shown in the panels - a single cell or an LCV? Please indicate in th e legend.

Figure 2D – what is shown in fist lane?

Figure3A – very low resolution, not clear what the red dot (arrow?) points to. What is depicted in the panels? Multiple cells? To my eye there is little difference between the panels. What is EV, please indicate.

Figure 3D – what is in first lane?

Figure 4C – what is shown in the panels, full cells or LCVs?

Figure 5C – same question as above

Figure 5E – no indication of expression of RavZLP.

Figure 6A – what are the different bands in the gel image? Please mark them In the last 5 lanes there is a strong band at ~25 kDa that appears to correspond to GFP but free GFP is only indicated in the first of these lanes. What is the band around 30 kDa in the first five lanes? There are double bands in the last five lanes where we would expect RavZ that seem to be interpreted as RavZLP and RavZLLO but they should not be in the same lanes.

Figure 7A, 8A – as above

Reviewer #2: This manuscript provides a comprehensive exploration of the role of the Dot/Icm effector RavZ during Legionella longbeachae infection. The L. pneumophila homologue is a cysteine protease that hydrolyzes lipidated LC3 to block autophagic maturation. Here the researchers demonstrate that the longbeachae homologue is also an effector and likely functions in the same way to suppress autophagy during infection. Interestingly the abundance of polyubiquitinated species on the LCV appears to be regulated by RavZ LLO but not RavZ LP indicating the possibility of a species specific function of this protein.

The study is comprehensive and scientifically sound and therefore entirely appropriate for publication in PLoS One. I have just a few minor suggestions:

* line 65-66: while flagellin is a key difference between longbeachae and pneumophila species it doesn't seem to account for the different infection outcomes in mice (consider Massis et al - J Infect Dis - where a pneumophila flagellin mutant still has different outcomes to longbeachae.

* line 81: "This apparatus comprises 25 proteins" - this is outdated given new publications Sheedlo et al and a couple of others

* Figure 3A - the quality of these IF images is unacceptable - it is impossible to determine any features within these images - they need to be higher resolution and perhaps also show a zoomed in section for the reader to deliniate LC3 puncta.

6. PLOS authors have the option to publish the peer review history of their article (what does this mean?). If published, this will include your full peer review and any attached files.

Reviewer #1: No

Reviewer #2: No

---

## [Author Response · Author response to Decision Letter 0]

18 Jan 2023

Authors’ responses to comments

Responses to Reviewer #1:

The manuscript describes the functional characterization of RavZ protein from Legionella longbeachae (RavZLLO) and its comparison with the RavZ ortholog from L. pneumophila (RavZLP). The authors demonstrate that RavZLLO is a secreted effector and interfere with host autophagy by removing the lipid anchor from LC3 protein, similarly to RavZLP. Moreover, RavZLLO can substitute for RavZLP in L. pneumophila. Next, the authors show that presence in the human cell RavZLLO but not RavZLP, either through direct expression or by infection with bacteria, decreases the level of polyubiquitination and polyneddylation and hypothesize that RavZLLO has deubiquitination activity. They support this view by noting that the N-terminal domain of RavZ displays fold similarity to Ubl-specific cysteine proteases, including well characterized Salmonella Typhimurium SseL.

1. While they observe decrease in the polyubiquitinated proteins in the human cell when RavZLLO is present, the same effect is caused by the mutant in which the active site Cys251 is replaced by Ala. In the alignment with SseL, Cys251 corresponds to active site cysteine in SseL. Moreover, no activity on various di-ubiquitins was demonstrated in vitro. The authors suggests that an association with an unknown host protein activated its deubiquitination activity. This is hand waving argument, and a more likely scenario is that RavZLLO stimulated deubiquitinase activity of another protein. That seems to me more reasonable explanation since RavZLP does not show this behavior, has closely similar N-terminal domain and differs in the C-terminal segment.

Response: Thanks for your suggestion. We have made the following revision. Lines 512-516: Therefore, it is possible that RavZLLO requires a cofactor to activate its DUB activity, as it is ectopically expressed in mammalian cells. Alternatively, RavZLLO might activate the DUB activity of host proteins to decrease cellular polyubiquitination and polyneddylation.

2. My other comment is to the discussion section. The first few pages repeat some of the introduction and are not relevant to the presented results, which are only discussed on pp.18-19.

Response: Thanks for your valuable suggestions. We have deleted the first two paragraph of the discussion section.

Figures:

3. Figure 2B – what is shown in the panels - a single cell or an LCV? Please indicate in the legend.

Response: Thanks for your comments. Figure 2B are the representative images showing the association of RavZLLO with the LCV. The LCVs formed in the infected cells were immunostained by anti-L. longbeachae antibodies. We have made the following revision. Lines 364-365: Representative LCVs with positive-association of RavZLLO during L. longbeachae infection.

4. Figure 2D – what is shown in fist lane?

Response: Thanks for the comment. The first lane shows the molecular weight marker (1610363, Bio-rad) in kDa.

5. Figure3A – very low resolution, not clear what the red dot (arrow?) points to. What is depicted in the panels? Multiple cells? To my eye there is little difference between the panels. What is EV, please indicate.

Response: Thanks for your advice. We apologized for the low resolution of the figures. And we have regenerated all figures at higher resolution which we hope to meet the standard. The red arrows indicate LC3 puncta (dots within the cells). The images were taken to show the effect of ectopically expressed RavZ on host autophagic responses which are represented by the formation of LC3 puncta. These images were taken to allow the inspection of multiple cells. In in cells transfected with empty vector, we can see multiple cells possessing LC3 puncta (dots) in the visual field. While in cells expressing RavZLP and RavZLLO, the LC3 puncta was barely detected. EV indicates empty vector, and we have descripted EV in the revised manuscript. Please see the line 407.

As also suggested by another reviewer, we have zoomed in the dots to see clear LC3 puncta. Please see our revised Fig 3A.

6. Figure 3D – what is in first lane?

Response: Thank you for your guidance. The first lane shows the molecular weight marker (1610363, Bio-rad) in kDa.

7. Figure 4C – what is shown in the panels, full cells or LCVs?

8. Figure 5C – same question as above

Response for Q7 and Q8: Thanks. Figure 4C and Figure 5C showed the A549 cells successfully infected with Legionella strains. Actually, these panels show both the cell and LCV. The LCVs were immunostained with anti-Legionella antibodies, whereas the cells were displayed by the expression of mCherry-LC3. These representative images were used to indicate the alteration in LC3 puncta formation after infected with Legionella.

9. Figure 5E – no indication of expression of RavZLP.

Response: Thanks for your careful review. In Fig 5E, the expression of RavZLP in L. pneumophila strains was determined by Western blot with the RavZLLO-specific antibody. It is possible that RavZLLO-specific antibody cannot recognize RavZLP. Since both RavZLLO and RavZLP genes were inserted into pZL507 which contains an N-terminal His tag in the plasmid. We further probed the expression of RavZLP and RavZLLO by probing the bacterial lysates with anti-His antibodies. As shown in the following Re-fig 1, RavZLP is indeed expressed in this strain. We have added this blot into revised Fig 5E.

Re-fig 1. The expression of RavZLP in L. pneumophila strains was determined by Western blot with the anti-His antibody.

10. Figure 6A – what are the different bands in the gel image? Please mark them In the last 5 lanes there is a strong band at ~25 kDa that appears to correspond to GFP but free GFP is only indicated in the first of these lanes. What is the band around 30 kDa in the first five lanes? There are double bands in the last five lanes where we would expect RavZ that seem to be interpreted as RavZLP and RavZLLO but they should not be in the same lanes.

Response: Thanks. We guess the ~25 kDa bands you have mentioned is on the left gel of the Fig 6A. The last 5 lanes are actually anti-HA IP products. The left gel was blotted with anti-HA antibody. Therefore, the ~25 kDa bands in the last 5 lanes are actually light chain released from the anti-HA agarose and recognized by the anti-HA antibody. We have marked the left gels in our revised Fig 6A. As to the 30 kDa bands shown in the first five lanes, we believed that they are protein in the cell lysates non-specifically recognized by anti-HA antibody. We guess the double bands in the last five lanes you mentioned are the bands around ~100 kDa of the left gel. This is anti-HA blot, therefore, GFP-tagged RavZ should not be shown in this gel. One explanation for these bands is that they are unknown proteins conjugated with HA-Ub. We have revised this figure to mark the heavy chain and light chain.

11. Figure 7A, 8A – as above

Response: Thanks for your comment. Fig 7A and Fig 8A are representative immunofluorescence images showing positive-association of ubiquitinated species with the LCVs. We have revised the figure legends of Fig 7A and Fig 8A. Lines 557-560: (A) Representative images of polyubiquitinated proteins associated with LCVs. U937 cells infected for 2 h with the indicated L. longbeachae strains were sequentially immunostained with antibodies specific for L. longbeachae and polyubiquitin (FK1). Lines 568-569: Representative images of polyubiquitinated proteins associated with LCVs. U937 cells were challenged with WT L. pneumophila, dotA- mutant, ∆ravZ, or ∆ravZ complemented with RavZLP/RavZLLO for 2 h.

Responses to Reviewer #2:

This manuscript provides a comprehensive exploration of the role of the Dot/Icm effector RavZ during Legionella longbeachae infection. The L. pneumophila homologue is a cysteine protease that hydrolyzes lipidated LC3 to block autophagic maturation. Here the researchers demonstrate that the longbeachae homologue is also an effector and likely functions in the same way to suppress autophagy during infection. Interestingly the abundance of polyubiquitinated species on the LCV appears to be regulated by RavZLLO but not RavZLP indicating the possibility of a species specific function of this protein.

The study is comprehensive and scientifically sound and therefore entirely appropriate for publication in PLoS One. I have just a few minor suggestions:

1. line 65-66: while flagellin is a key difference between longbeachae and pneumophila species it doesn't seem to account for the different infection outcomes in mice (consider Massis et al - J Infect Dis - where a pneumophila flagellin mutant still has different outcomes to longbeachae.

Response: Thanks for your careful review and valuable comment. We also read the J infect Dis paper by Massis et al. One of the major differences between L. pneumophila and L. longbeachae is their ability to colonize the lungs of mice. While only A/J mice are permissive for replication of L. pneumophila, A/J, C57BL/6 and BALB/c mice are all permissive for replication of L. longbeachae [1, 2]. Resistance of C57BL/6 and BALB/c mice to L. pneumophila has been attributed to polymorphisms in Nod-like receptor apoptosis inhibitory protein 5 (naip5) allele [3, 4]. The current model states that L. pneumophila replication is restricted due to flagellin dependent caspase-1 activation through Naip5-Ipaf and early macrophage cell death by pyroptosis. Later on, the genomic sequencing analyses reveal that L. longbeachae does not encode flagella biosynthesis genes, thereby providing a possible explanation for differences in mouse susceptibility to infection between the two species [5, 6]. In Massis’s study, they found that L. longbeachae is more virulent than L. pneumophila strain lacking flagellin in a mouse model of infection, the reduced activation of the innate immunity and reduced induction of cytokines during L. longbeachae infection may contribute to increased bacterial colonization of the lungs, replication, and dissemination, a process that may be critical for the pathogenesis and induction of death in the infected individuals.

We have revised this sentence. Lines 63-65: “Unlike L. pneumophila, L. longbeachae does not encode flagella biosynthesis genes, thus partly explaining the differences in mouse susceptibility between these two species”.

2. line 81: "This apparatus comprises 25 proteins" - this is outdated given new publications Sheedlo et al and a couple of others

Response: Thanks for your valuable suggestions. We have revised our manuscript according to Sheedlo et al. “This apparatus comprises 27 proteins and injects a set of effector proteins across both bacterial and vacuole membranes into the cytosol of infected host cells or the vacuole lumen.” Please see the lines 81-83 in the revised manuscript.

3. Figure 3A - the quality of these IF images is unacceptable - it is impossible to determine any features within these images - they need to be higher resolution and perhaps also show a zoomed in section for the reader to deliniate LC3 puncta.

Response: Thanks for your advice. We apologized for the low resolution of the figures. And we have regenerated all figures at higher resolution which we hope to meet the standard. The images were taken to show the effect of ectopically expressed RavZ on host autophagic responses which are represented by the formation of LC3 puncta. In cells transfected with empty vector, we can see multiple cells possessing LC3 puncta (dots) in the visual field. While in cells expressing RavZLP and RavZLLO, the LC3 puncta was barely detected. We have zoomed in the dot regions as your suggestion. Please see our revised Fig 3A.

References

1. Asare R, Santic M, Gobin I, Doric M, Suttles J, Graham JE, et al. Genetic susceptibility and caspase activation in mouse and human macrophages are distinct for Legionella longbeachae and L. pneumophila. Infection and immunity. 2007;75(4):1933-45 https://doi.org/10.1128/iai.00025-07. PMID: 17261610.

2. Gobin I, Susa M, Begic G, Hartland EL, Doric M. Experimental Legionella longbeachae infection in intratracheally inoculated mice. Journal of medical microbiology. 2009;58(Pt 6):723-30 https://doi.org/10.1099/jmm.0.007476-0. PMID: 19429747.

3. Molofsky AB, Byrne BG, Whitfield NN, Madigan CA, Fuse ET, Tateda K, et al. Cytosolic recognition of flagellin by mouse macrophages restricts Legionella pneumophila infection. The Journal of experimental medicine. 2006;203(4):1093-104 https://doi.org/10.1084/jem.20051659. PMID: 16606669.

4. Wright EK, Goodart SA, Growney JD, Hadinoto V, Endrizzi MG, Long EM, et al. Naip5 affects host susceptibility to the intracellular pathogen Legionella pneumophila. Current biology : CB. 2003;13(1):27-36 https://doi.org/10.1016/s0960-9822(02)01359-3. PMID: 12526741.

5. Cazalet C, Gomez-Valero L, Rusniok C, Lomma M, Dervins-Ravault D, Newton HJ, et al. Analysis of the Legionella longbeachae genome and transcriptome uncovers unique strategies to cause Legionnaires' disease. PLoS genetics. 2010;6(2):e1000851 https://doi.org/10.1371/journal.pgen.1000851. PMID: 20174605.

6. Kozak NA, Buss M, Lucas CE, Frace M, Govil D, Travis T, et al. Virulence factors encoded by Legionella longbeachae identified on the basis of the genome sequence analysis of clinical isolate D-4968. Journal of bacteriology. 2010;192(4):1030-44 https://doi.org/10.1128/jb.01272-09. PMID: 20008069.

---

## [Decision Letter · Decision Letter 1]

27 Jan 2023

Legionella longbeachae effector protein RavZ inhibits autophagy and regulates phagosome ubiquitination during infection

PONE-D-22-30067R1

Dear Dr. Qiu,

We’re pleased to inform you that your manuscript has been judged scientifically suitable for publication and will be formally accepted for publication once it meets all outstanding technical requirements.

Kind regards,

Aristóbolo M Silva

Academic Editor

PLOS ONE

Additional Editor Comments (optional):

Reviewers' comments:

Reviewer's Responses to Questions

**Comments to the Author**

1. If the authors have adequately addressed your comments raised in a previous round of review and you feel that this manuscript is now acceptable for publication, you may indicate that here to bypass the “Comments to the Author” section, enter your conflict of interest statement in the “Confidential to Editor” section, and submit your "Accept" recommendation.

Reviewer #1: All comments have been addressed

Reviewer #2: All comments have been addressed

2. Is the manuscript technically sound, and do the data support the conclusions?

Reviewer #1: Yes

Reviewer #2: (No Response)

3. Has the statistical analysis been performed appropriately and rigorously? 

Reviewer #1: Yes

Reviewer #2: (No Response)

4. Have the authors made all data underlying the findings in their manuscript fully available?

Reviewer #1: Yes

Reviewer #2: (No Response)

5. Is the manuscript presented in an intelligible fashion and written in standard English?

Reviewer #1: Yes

Reviewer #2: (No Response)

6. Review Comments to the Author

Reviewer #1: The authors responded in an adequate manner to my comments. The changes have been made in the text of the manuscript.

Reviewer #2: (No Response)

7. PLOS authors have the option to publish the peer review history of their article (what does this mean?). If published, this will include your full peer review and any attached files.

Reviewer #1: No

Reviewer #2: No

---

## [Editor Report · Acceptance letter]

1 Feb 2023

PONE-D-22-30067R1 

*Legionella longbeachae* effector protein RavZ inhibits autophagy and regulates phagosome ubiquitination during infection 

Dear Dr. Qiu:

I'm pleased to inform you that your manuscript has been deemed suitable for publication in PLOS ONE. Congratulations! Your manuscript is now with our production department. 

Kind regards, 

on behalf of

Dr. Aristóbolo M Silva 

Academic Editor

PLOS ONE